Phosphite effects on sugarcane growth and biochemicals under in vitro osmotic stress

Martínez-Ballesteros Jennifer 1
Bañuelos-Hernández Karina P. 1
Rodríguez-Lagunes Daniel A. 1
Hidalgo-Contreras Juan V. 2
Pastelín-Solano Miriam C. 3
Vivar-Vera Guadalupe 3
Bulbarela-Marini Javier E. 3
Castañeda-Castro Odon odcastaneda@uv.mx 3
1 Faculty of Biological and Agricultural Sciences, University of Veracruz , Amatlán de los Reyes , Veracruz , Mexico
2 College of Postgraduates in Agricultural Sciences campus Córdoba , Amatlán de los Reyes , Veracruz , Mexico
3 Faculty of Chemical Sciences, University of Veracruz , Orizaba , Veracruz , Mexico
Domingues Douglas
Electronic publication date: 2025 Aug 5
Publication date: 2025
Volume: 13
Electronic Location ID: e19763
Received 2025 Apr 16; Accepted 2025 Jun 26
Copyright: ©2025 Martínez-Ballesteros et al.
Copyright year: 2025
Copyright holder: Martínez-Ballesteros et al.
License: This is an open access article distributed under the terms of the Creative Commons Attribution License, which permits unrestricted use, distribution, reproduction and adaptation in any medium and for any purpose provided that it is properly attributed. For attribution, the original author(s), title, publication source (PeerJ) and either DOI or URL of the article must be cited.
License URL: https://creativecommons.org/licenses/by/4.0/

Keywords: Saccharum spp., Phosphorous acid, Beneficial elements, Oxyanion

Funding: Secretariat of Science, Humanities, Technology, and Innovation (SECIHTI) This study was funded by the Secretariat of Science, Humanities, Technology, and Innovation (SECIHTI). There was no additional external funding received for this study. The funders had no role in study design, data collection and analysis, decision to publish, or preparation of the manuscript.

==============================
Background

Biostimulants positively impact plant growth, yield, and chemical composition while enhancing tolerance to biotic and abiotic stress. Phosphite (Phi), a phosphate analog, has been proposed as a biostimulant due to its advantages over traditional phosphate fertilizers and herbicides.

Methods

This study evaluated the effects of Phi on sugarcane seedlings (CP 72-2086) under conventional (non-stress) and osmotic stress conditions during in vitro multiplication. Seedlings were treated with Phi at 0.1, 0.3, and 0.5 mM (derived from H3PO3) for 30 days, followed by 7 days of osmotic stress induced with 10% polyethylene glycol 6000 (PEG).

Results

Phi significantly increased leaf length, width, and number, as well as shoot count. Additionally, it enhanced foliar concentrations of chlorophylls a and b, sugars, and amino acids under both conventional and osmotic stress conditions. In conclusion, Phi serves as an effective inorganic biostimulant for sugarcane (CP 72-2086) during in vitro multiplication, stimulating seedling growth and modulating essential biomolecule concentrations.

Introduction

Sugarcane (Saccharum spp.) is a socioeconomically vital agro-industrial crop with high photosynthetic efficiency and sucrose storage capacity, making it a key source of carbohydrates for food and bioenergy (Bordonal et al., 2018; Aguilar-Rivera, 2019). The global population, currently estimated at 8.1 billion, is projected to increase by 34% over the next three decades, escalating demands for food and industrial sugar (Kumar et al., 2020). However, sugarcane cultivation faces critical challenges linked to climate change, including rising temperatures, intensified droughts, and biodiversity loss, which disrupt plant metabolism, induce morphological and physiological alterations, and reduce crop productivity (Gómez-Merino et al., 2022).

In Mexico, sugarcane cultivation plays a pivotal role in the national agro-industrial sector. Currently, 80% of the sugar consumed globally originates from this grass. Mexico’s total land area spans 198 million hectares, with only 27.5 million hectares dedicated to agriculture. Of these, approximately 800,000 hectares are annually cultivated with sugarcane (FAO, 2024). The Mexican sugar industry relies primarily on four commercial varieties: CP 72-2086, Mex 69-290, Mex 79-431, and ITV 92-1424, which occupy 31%, 26%, 8%, and 6% of the planted area, respectively. Among these, CP 72-2086 has dominated cultivation in recent years due to its adaptability and yield performance (Dias-Kanthack et al., 2020).

Sustainable improvement of sugarcane cultivars is imperative to maintain high sucrose yields under unpredictable environmental stressors, population growth, and resource constraints (Mansoori, Khayat & Jorphi, 2014). Current agricultural practices, reliant on excessive fertilizers, compromise soil health and food quality, underscoring the need for innovative solutions (Aguilar-Rivera et al., 2018).

Plant biotechnology, particularly micropropagation, offers advantages in producing genetically uniform and vigorous planting material, ensuring robust field establishment (Bello-Bello Mendoza-Mexicano & Pérez-Sato, 2018; Redae & Ambaye, 2018). Biostimulants, such as phosphite (Phi), have emerged as tools to enhance physiological vigor, nutrient assimilation, and abiotic stress tolerance. These compounds improve crop productivity and quality while activating defense mechanisms against environmental stressors (Han et al., 2021).

Phosphite (HPO3−), derived from phosphorous acid, functions as both a nutrient source and biostimulant. It promotes root development, nutrient uptake, and stress resilience in crops, including sugarcane (Gómez-Merino & Trejo-Téllez, 2015; Halpern et al., 2015). Phi has been utilized as a pesticide and growth enhancer in diverse agricultural systems, including turfgrass and field-grown sugarcane, where it improves morphological and biochemical markers during early growth stages (Gómez-Merino & Trejo-Téllez, 2015; Martínez-Ballesteros et al., 2024). In vitro cultures, biostimulants like Phi modulate plant metabolism, enhancing tissue growth and altering secondary metabolite levels (Berkowitz et al., 2013; Al-Mayahi, 2019).

Agricultural stress exerts detrimental effects on plant growth and metabolic processes, with many drought-induced changes reflecting general adaptive modulation under adverse conditions. Plant performance under stress hinges on the balance between its damaging impacts and the activation of protective mechanisms (Xiong et al., 2012). Water stress simulation using polyethylene glycol (PEG) has been widely adopted to induce osmotic stress in vitro, as it reduces the medium’s water potential without phytotoxic effects, enabling controlled study of drought impacts on growth and biochemical responses (Hamayun et al., 2010). Studies highlight phosphite (Phi) as a biostimulant that enhances antioxidant enzymatic activities, mitigating oxidative damage in plants (Raposo-Junior, Gomes-Neto & Silva-Sacramento, 2013). However, research on Phi’s efficacy under drought stress in sugarcane remains limited, particularly regarding its potential to sustain productivity under water-deficit conditions.

This study aimed to evaluate the effects of Phi on growth and biochemical parameters in sugarcane seedlings (CP 72-2086) under conventional (non-stress) and polyethylene glycol 6000 (PEG)-induced osmotic stress conditions during vitro multiplication.

Materials & Methods

Plant material and disinfection

The plant material used for the study consisted of stem tips containing the meristem of sugarcane varieties CP 72-2086, which were collected from the experimental field of the Faculty of Biological and Agricultural Sciences located in the municipality of Amatlán de los Reyes, Veracruz, Mexico. The apical meristems were treated with a 20% sodium hypochlorite solution (CLORALEX®) and 20 drops of Tween™ for 20 min. Finally, the explants were cultured on MS medium (Murashige & Skoog, 1962) for 30 days.

Vegetative growth

The biostimulant response of Phi (phosphite) was evaluated at doses of 0, 0.1, 0.3, and 0.5 mM, derived from 98% phosphorous acid (H3PO3; Sigma-Aldrich®; St. Louis, MO, USA). The Phi was incorporated into liquid MS medium (Murashige & Skoog, 1962), supplemented with ascorbic acid (100 mg L−1), citric acid (150 mg L−1), kinetin (one mg L−1), and sucrose (30 g L−1). The pH was adjusted to 5.7 using NaOH or HCl (0.1–1.0 N). The medium was sterilized at 120 °C for 15 min in a vertical autoclave (Lab-Tech LAC5060s; Namyangju, South Korea).

Twelve shoots, each two cm in length, were introduced into each Rita® bioreactor (Alvard, Cote & Teisson, 1993) containing 200 mL of MS culture medium and subjected to a 30-day cultivation period. The immersion time was 5 min, with frequencies of 4, 8, and 12 h. The photon flux density was maintained between 40 and 50 µmol m−2 s−1, with a photoperiod of 16 h of light provided by white fluorescent lamps (General Electric; Wayne, PA, USA) and 8 h of darkness. The temperature was maintained at 25 ± 2 °C during the day and 18 ± 2 °C at night.

After 30 days of biostimulation, half of the seedlings were evaluated for morphological and biochemical parameters. The other half were subjected to osmotic stress conditions induced by 10% polyethylene glycol 6000 (PEG), while maintaining the Phi doses (0, 0.1, 0.3, and 0.5 mM), for an additional 7 days to evaluate their morphological and biochemical parameters under these conditions. The osmotic potentials of the eight evaluated treatments were determined using a Vapro 5520 osmometer (Wescor; Logan, UT, USA). The treatments and their osmotic potentials were as follows: Control Phi 0 mM (−0.07 MPa), 10% PEG without Phi (−0.182 MPa), Phi 0.1 mM (−0.08 MPa), Phi 0.1 mM + 10% PEG (−0.196 MPa), Phi 0.3 mM + 10% PEG (−0.185 MPa), and Phi 0.5 mM + 10% PEG (−0.174 MPa).

After 30 days of Phi treatment (0, 0.1, 0.3, 0.5 mM) and 7 days with or without osmotic stress induced by PEG, the seedlings were removed from the MS medium and evaluated. The number of shoots and leaves, shoot and leaf length (cm), and leaf width (cm) were recorded using millimeter paper with a 0.1 cm scale, and the data were processed using the ImageJ software (https://imagej.net/ij/download.html) (Rueden et al., 2017). Fresh biomass weight was determined using an OHAUS analytical digital scale, model AV213 Adventurer Pro (Parsippany, NJ, USA). Dry biomass weight was determined using a forced-air circulation oven (HCF-125D; Riossa, Monterrey, Mexico) for 72 h at 70 °C.

Concentration of chlorophyll a, b, and total chlorophyll in leaves and stems

The quantification of chlorophyll a, b, and total chlorophyll was determined using the method described by Harborne (1973). Fresh leaf tissue (0.1 g) from each sample was placed in 80% acetone and macerated for 24 h at 4 °C. The mixture was filtered, and the supernatants from each extraction were collected and used for chlorophyll determination. The concentrations of chlorophyll a and b in leaves and stems were determined. The extracts were quantified at 635 and 645 nm using a UV/Vis spectrophotometer (Benchmark Scientific, Sayreville, NJ, USA). The total chlorophyll concentration and the chlorophyll a/b ratio were also calculated.

Concentration of total free amino acids in leaves and stems

The concentration of total free amino acids in plant tissues was determined using the ninhydrin method (Moore & Stein, 1954; modified by Sun et al., 2006). Samples were placed in 500 µL of 80% ethanol and incubated in a water bath (Benchmark Scientific) at 80 °C for 20 min. Then, 250 µL of the supernatant was mixed with 250 µL of a sodium citrate (16 mM)/ascorbic acid (34 mM) solution (pH 5.2) and 500 µL of ninhydrin (1% w/v; Sigma-Aldrich, Steinheim, Germany) in 70% ethanol (v/v). After incubating at 95 °C for 20 min, the samples were cooled to room temperature. Absorbance was measured at 570 nm using a UV/Vis spectrophotometer (Benchmark Scientific). A calibration curve was prepared using leucine (Sigma-Aldrich). Four replicates were prepared for each treatment, with two replicates each.

Concentration of total soluble sugars in leaves

The foliar concentration of total soluble sugars was determined using the anthrone method (Brummer & Cui, 2005) (based on Southgate, 1976). Samples were placed in 40 mL of 80% ethanol (v/v) and incubated in a water bath (Benchmark Scientific) at 125 °C until complete evaporation. The precipitate was resuspended in 20 mL of distilled water, and a 500 µL aliquot was mixed with 500 µL of 80% ethanol (v/v). Then, five mL of anthrone (Sigma-Aldrich) was added. The samples were transferred to a water bath at 95 °C for 15 min and then placed on ice. The sugar concentration was determined using a standard curve prepared with glucose (Sigma-Aldrich). Absorbance was measured at 620 nm using a UV/Vis spectrophotometer (Benchmark Scientific).

Concentration of proline in shoots

The concentration of proline in plant tissues was determined using 0.5 g of shoot and one g of root. The tissues were macerated with 3% sulfosalicylic acid (w/v) and filtered using Whatman No. 2 filter paper. Then, two mL of the extract was mixed with two mL of acidic ninhydrin and two mL of glacial acetic acid and incubated at 100 °C for 60 min. After cooling, four mL of toluene (JT Baker) was added to each sample. Absorbance was measured at 570 nm using a UV/Vis spectrophotometer (Benchmark Scientific). The proline concentration was determined using a standard curve prepared with L-proline.

Analysis of plant hormones

For the determination of phytohormones, a protocol based on high-performance liquid chromatography (HPLC) was employed, optimizing extraction and quantification conditions to ensure high sensitivity and reproducibility (Pan, Welti & Wang, 2010). Plant tissue was ground in liquid nitrogen to obtain a fine powder. Then, 50 mg of each sample was weighed and mixed with 500 µL of extraction solvent (2-propanol/H2O/concentrated HCl, 2:1:0.002, v/v/v). After adding one mL of dichloromethane, the samples were shaken for 30 min and kept at 4 °C during solvent extraction. Approximately 900 µL of the supernatant was transferred, dissolved in 0.1 mL of methanol, and 50 µL of the sample solution was injected into the HPLC column. Quantification was performed using HPLC-UV/VIS (model ICS3000; Dionex, San Jose, CA, USA).

Statistical analysis

A completely randomized design with a factorial arrangement was applied to the treatments. The first fixed factor corresponded to phosphite (Phi) concentrations (0, 0.1, 0.3, and 0.5 mM), evaluated under a second fixed factor labeled PEG, which represented conventional conditions and osmotic stress induced by 10% polyethylene glycol (PEG). Each treatment was replicated four times, with individual seedlings serving as experimental units. Twelve shoots were placed per bioreactor.

The assumptions of normality (Shapiro–Wilk test, α = 0.05) and homogeneity of variance (Levene’s test, α = 0.05) were verified. Subsequently, a two-way analysis of variance (ANOVA) with interaction was performed to assess the effects of phosphite (Phi), osmotic stress conditions (PEG), and their interaction (Phi × PEG). Treatment means were compared using Tukey’s test (P ≤ 0.05). All statistical analyses were conducted in R version 4.4.1 (R Core Team, 2024).

A principal component analysis (PCA) was performed to evaluate the variability of morphological and biochemical responses of sugarcane seedlings based on phosphite (Phi) concentration and the presence of osmotic stress induced by 10% polyethylene glycol 6000 (PEG). The analysis was carried out using the FactoMineR package in R (version 4.4.1; R Core Team, 2024), with variables standardized using z-score transformation to ensure comparable variance among parameters. The first two principal components (PC1 and PC2) were considered for interpreting variability, and a biplot was used to visualize the correlation between treatments and measured variables.

A heatmap was constructed to illustrate differential treatment responses across biochemical and morphological parameters. The data matrix was column-normalized and visualized using the pheatmap() function (Kolde, 2019) in R, with hierarchical clustering (Euclidean distance, Ward’s linkage method). A divergent color gradient (blue = low values, red = high values) represented response intensities across treatments.

Results

Phosphite (Phi) influences the vegetative growth of sugarcane seedlings under stress conditions

The application of Phi at doses of 0.1, 0.3, and 0.5 mM for 30 days (Figs. 1B, 1C, 1D) had positive effects on the growth of sugarcane seedlings (CP 72-2086), increasing height, shoot number, and biomass compared to the control treatment (Phi 0 mM). Treatments with Phi at 0.3 mM and 0.5 mM, followed by the addition of 10% PEG, maintained their biostimulated morphological characteristics, showing superior performance compared to seedlings treated with Phi 0 mM + 10% PEG and Phi 0.1 mM + 10% PEG.

Figure 1 Sugarcane CP 72-2086 growth under phosphite (0–0.5 mM) ±10% PEG stress, showing dose-dependent morphological changes.

(A) Control without Phi or PEG (0 mM); B) Phi 0.1 mM; (C) Phi 0.3 mM; (D) Phi 0.5 mM; (E) control without Phi + 10% PEG; (F) Phi 0.1 mM + 10% PEG; (G) Phi 0.3 mM + 10% PEG; (H) Phi 0.5 mM + 10% PEG. The images show differences in morphological development under the evaluated conditions.

The application of Phi at doses of 0.1, 0.3, and 0.5 mM for 30 days (Figs. 2A–2E) significantly promoted the morphological characteristics of sugarcane seedlings (CP 72-2086), such as the number and length of shoots, as well as the number, length, and width of leaves. Among these, the dose of 0.3 mM Phi showed the greatest biostimulant effect in the absence of PEG, increasing the number of shoots (13.6 ± 0.8, Fig. 2A) and shoot length (1.4 ± 0.08 cm, Fig. 2B) compared to the control (4.3 ± 0.3 shoots and 0.6 ± 0.04 cm, respectively). Additionally, this dose also stood out in the number of leaves (26.5 ± 1.0, Fig. 2C), leaf length (12.2 ± 0.9 cm, Fig. 2D), and leaf width (0.7 ± 0.02 cm, Fig. 2E), outperforming the other evaluated doses.

Figure 2 Variation in the morphological characteristics of sugarcane seedlings (CP 72-2086) treated with different phosphite (Phi) concentrations and the presence of osmotic stress induced by 10% PEG.

(A) Number of shoots, (B) shoot length, (C) number of leaves, (D) leaf length, and (E) leaf width. The treatments included: control (Phi 0 mM), 10% PEG without Phi, Phi 0.1 mM, Phi 0.1 mM + 10% PEG, Phi 0.3 mM + 10% PEG, and Phi 0.5 mM + 10% PEG. Mean values are displayed on each bar. Different letters indicate statistically significant differences between treatments (Tukey, P ≤ 0.05).

Under osmotic stress conditions induced by 10% PEG, seedlings treated with 0.3 mM Phi showed notable resistance, maintaining higher values in all morphological variables compared to the control with 10% PEG. For example, the number of shoots increased by 12.06%, while shoot length increased by 16.6% compared to the average of the other PEG treatments (Figs. 2A–2B).

Regarding biomass, the fresh biomass weight of shoots was significantly higher in seedlings treated with 0.3 mM Phi (2.22 ± 0.09 g) without PEG compared to the control (1.2 ± 0.04 g, Fig. 3A). Under PEG-induced stress, the same dose (0.3 mM Phi) increased fresh biomass to 1.8 ± 0.04 g. Similarly, the dry biomass weight of shoots followed a similar pattern, reaching 0.40 ± 0.01 g with 0.3 mM Phi in the absence of PEG and 0.40 ± 0.02 g in the presence of 10% PEG, significantly outperforming the control and the treatment with PEG without Phi (Fig. 3B).

Figure 3 Biomass weight of sugarcane seedlings (CP 72-2086) treated with phosphite (Phi) and osmotic stress induced by 10% PEG.

(A) Fresh biomass weight and (B) dry biomass weight. The treatments included: control (Phi 0 mM), 10% PEG without Phi, Phi 0.1 mM, Phi 0.1 mM + 10% PEG, Phi 0.3 mM + 10% PEG, and Phi 0.5 mM + 10% PEG. Mean values are displayed on each bar. Different letters indicate statistically significant differences between treatments (Tukey, P ≤ 0.05).

Phosphite (Phi) influences chlorophyll concentration in sugarcane seedlings (CP 72-2086) under stress conditions

The 0.3 mM Phi dose without PEG showed the highest averages in chlorophyll a, chlorophyll b, and total chlorophyll concentrations, with values of 1.33 ± 0.07 mg g−1, 0.49 ± 0.02 mg g−1, and 1.81 ± 0.09 mg g−1, respectively, compared to the control without PEG, which had values of 0.47 ± 0.03 mg g−1, 0.31 ± 0.01 mg g−1, and 0.78 ± 0.03 mg g−1 (Figs. 4A, 4B, and 4C). Similarly, shoots treated with 0.3 mM Phi and exposed to 10% PEG significantly increased chlorophyll a (1.17 ± 0.06 mg g−1), chlorophyll b (0.46 ± 0.013 mg g−1), and total chlorophyll (1.63 ± 0.05 mg g−1) compared to the control in the presence of PEG (0.24 ± 0.01 mg g−1, 0.19 ± 0.01 mg g−1, and 0.44 ± 0.02 mg g−1, respectively). It is worth noting that the remaining treatments also showed significantly higher values than the control in all evaluated variables (Figs. 4A, 4B, and 4C). On the other hand, the chlorophyll a/b ratio reached its highest value in the 0.1 mM Phi treatment without PEG (4.01 ± 0.4), followed by the 0.3 mM Phi treatment without PEG (2.84 ± 0.18), compared to the control without PEG (1.37 ± 0.11) (Fig. 4D).

Figure 4 Chlorophyll a (A), b (B), total (C), and a/b ratio (D) in sugarcane CP 72-2086 treated with phosphite (0–0.5 mM) ±10% PEG.

The treatments included: Control (Phi 0 mM), 10% PEG without Phi, Phi 0.1 mM, Phi 0.1 mM + 10% PEG, Phi 0.3 mM + 10% PEG, and Phi 0.5 mM + 10% PEG. Mean values are displayed on each bar. Different lowercase letters indicate statistically significant differences between treatments (Tukey, P ≤ 0.05).

Phosphite (Phi) influences the concentration of free amino acids and total sugars in sugarcane seedlings (CP 72-2086) under stress conditions

The treatment with 0.5 mM Phi without PEG increased the concentration of total free amino acids in the shoots to an average of 0.30 ± 0.01 nM g−1 compared to the control without Phi, which showed an average of 0.07 ± 0.004 nM g−1. However, the same dose of Phi in the presence of 10% PEG reduced the concentration to 0.16 ± 0.006 nM g−1, although it maintained statistically significant differences compared to the control with PEG, whose average value was 0.03 ± 0.004 nM g−1 (Fig. 5A).

Figure 5 Concentration of total free amino acids (A) and total soluble sugars (B) in sugarcane seedlings treated with different doses of Phi in the presence or absence of 10% PEG.

The treatments included: control (Phi 0 mM), 10% PEG without Phi, Phi 0.1 mM, Phi 0.1 mM + 10% PEG, Phi 0.3 mM + 10% PEG, and Phi 0.5 mM + 10% PEG. Mean values are displayed on each bar. Different letters indicate statistically significant differences between treatments (Tukey, P ≤ 0.05).

The concentration of total soluble sugars significantly increased in the treatments with 0.1, 0.3 and 0.5 mM Phi without PEG, with averages of 0.20 ± 0.01 mg g−1, 0.26 ± 0.01 mg g−1, and 0.21 ± 0.008 mg g−1, respectively, compared to the control without Phi (0.12 ± 0.006 mg g−1). On the other hand, the 0.3 mM Phi dose in the presence of 10% PEG showed an average sugar concentration of 0.20 ± 0.009 mg g−1, higher than the control with PEG, whose average was 0.08 ± 0.005 mg g−1 (Fig. 5B).

Phosphite (Phi) influences proline concentration in sugarcane seedlings (CP 72-2086) under stress conditions

Proline concentration exhibited a significant increase in the 0.3 mM Phi treatment without PEG, reaching an average of 0.61 ± 0.006 µM g−1 FW, compared to the PEG-free control (0.20 ± 0.01 µM g−1 FW). Under 10% PEG stress, the highest value was also observed in the 0.3 mM Phi treatment, with an average concentration of 0.39 ± 0.019 µM g−1 FW, surpassing the PEG-stressed control (0.16 ± 0.007 µM g−1 FW) (Fig. 6).

Figure 6 Proline concentration in sugarcane seedlings treated with different concentrations of Phi and 10% PEG.

Mean values are displayed on each bar. Different letters indicate statistically significant differences between treatments (Tukey, P ≤ 0.5).

Figure 7 Amount of abscisic acid in sugarcane seedlings treated with different concentrations of Phi and 10% PEG.

Mean values are displayed on each bar. Different letters indicate statistically significant differences between treatments (Tukey, P ≤ 0.05).

Figure 8 Principal component analysis (PCA) of morphological and biochemical variables in sugarcane seedlings (CP 72-2086) treated with Phi and subjected to osmotic stress with 10% PEG.

The graph represents the distribution of treatments based on the first two principal components (PC1 and PC2). The arrows indicate the contribution of each variable to the model, where longer vectors represent a greater influence on the explained variability.

Phosphite (Phi) influences the concentration of phytohormones in sugarcane seedlings (CP 72-2086) under stress conditions

The amount of abscisic acid (ABA) showed a significant increase in the 0.3 mM Phi treatment with PEG, with an average of 28.2 ± 1.91 ng/g, which was higher than the amount in the control without and with PEG. Similarly, the other treatments with Phi and Phi + 10% PEG also showed significant differences compared to the control (Fig. 7).

The PCA revealed a clear separation between treatments based on phosphite (Phi) concentration and the presence of 10% PEG-induced osmotic stress (Fig. 8). Treatments with 0.3 mM Phi under PEG-free conditions clustered in the upper right quadrant, showing strong correlations with increased shoot number, higher fresh biomass, and elevated chlorophyll and proline concentrations. In contrast, Phi-free treatments exposed to 10% PEG were positioned in the lower left quadrant, indicating a pronounced association with reduced growth and secondary metabolism. Two-way ANOVA confirmed significant effects of Phi concentration on nearly all evaluated variables (P ≤ 0.05; Tables 1 and 2), with the Phi × osmotic stress interaction also significantly influencing key parameters such as proline accumulation, chlorophyll content, and shoot biomass (P ≤ 0.05; Tables 1 and 2). Principal component 1 (PC1) accounted for 75.7% of the total variability, while PC2 explained 8.9%. These results highlight that Phi-mediated modulation of plant metabolism significantly influences the physiological and biochemical responses of sugarcane seedlings under both optimal and stress conditions.

The representation of the data using a heatmap revealed a differentiated pattern in the accumulation of metabolites and morphological development in response to Phi and PEG (Fig. 9). Treatments with 0.3 mM and 0.5 mM Phi without PEG showed higher concentrations of total chlorophyll, proline, and soluble sugars, while treatments without Phi and exposed to PEG exhibited a reduction in these parameters. However, the combination of Phi and PEG maintained an increase in ABA. Additionally, hierarchical analysis grouped treatments with 0.3 mM and 0.5 mM Phi within the same cluster, reflecting similarities in their biostimulant effect, while controls and treatments with PEG without Phi formed a separate cluster, indicating a differential response to osmotic stress.

Discussion

In this study, treatments with Phi demonstrated a beneficial effect as a biostimulant derived from H3PO3 during the micropropagation of sugarcane (variety CP 72-2086). The evaluated doses of Phi revealed a significant interaction with morphometric parameters, such as the number and length of shoots and leaves, leaf width, and fresh and dry biomass weight. This suggests a positive influence of the phosphite molecule on this specific variety.

Biostimulants trigger hormetic responses, a phenomenon characterized by stimulation at low doses and inhibitory or toxic effects at high concentrations (Jalal et al., 2021). In the case of Phi, its efficacy depends on both the appropriate dose and the source of the biostimulant; inadequate concentrations or doses can reduce growth, decrease leaf area, and lead to phosphorus uptake deficits (Exley, 2015; Vinas, Méndez & Jiménez, 2020).

Morphological measurements, such as height, diameter, and biomass, along with biochemical parameters like chlorophyll, amino acids, soluble sugars, proline, and proteins, are critical determinants of seedling quality and their ability to adapt to field conditions, which is crucial for the commercial productivity of crops like sugarcane (Pessoa et al., 2019). The favorable growth in morphometric variables could be attributed to the rapid absorption and translocation of Phi from the roots to meristematic tissues, facilitated by transport through the xylem (Ávila et al., 2013).

The increase in fresh and dry biomass weight in this study is also related to the effects of Phi on photosynthesis, including enhanced photosynthetic activity, hydraulic conductance, and the expression of aquaporins, which are fundamental under water stress conditions (Akram et al., 2018).

Table 1 Significance analysis of the effect of phosphite doses, PEG and their interaction (phosphite × PEG) on growth indicators of the CP 72-2086 sugarcane variety in vitro.

Factors	Number of shoots	Shoot length
(cm)	Number of leaves	Leaves length
(cm)	Leaves width
(cm)	Fresh biomass weight (g)	Dry biomass weight (g)	
Phosphite	0.001	0.001	0.001	0.001	0.001	0.001	0.001	
PEG	0.001	0.01	0.001	0.1	0.01	0.001	0.001	
Phosphite × PEG	0.05	0.001	0.1	0.001	0.001	0.01	0.001	
Notes.

Significant difference at P ≤ 0.05. No significant difference (ns).

Table 2 Significance analysis of the effect of doses of phosphite, PEG and their interaction (phosphite × PEG) on biochemical indicators of sugarcane variety CP 72–2086 in vitro.

Factors	Chlorophyll a	Chlorophyll b	Chlorophyll total	Chlorophyll a/b	Free ammino acids nM g−1	Total soluble sugars mg g−1	Proline
µM g−1	ABA
ng/g	
	mg g−1					
Phosphite	0.001	0.001	0.001	0.001	0.001	0.001	0.001	0.001	
PEG	0.001	0.01	0.001	0.001	0.001	0.001	0.001	0.001	
Phosphite × PEG	0.05	0.001	0.1	0.001	0.001	0.1	0.001	0.05	
Notes.

Significant difference at P ≤ 0.05. No significant difference (ns).

Under osmotic stress conditions, it was observed that 0.3 mM Phi significantly stimulated the recovery and growth of seedlings in the presence of 10% PEG, compared to plants without Phi under the same stress level. This aligns with studies showing that biostimulants at low concentrations increase tolerance to various types of abiotic stress (Coskun et al., 2019), a relevant finding for addressing drought events (Pandey & Shukla, 2015).

Figure 9 Heatmap of the biochemical and morphological responses of sugarcane seedlings (CP 72-2086) treated with phosphite (Phi) in the presence or absence of osmotic stress induced by 10% PEG.

Phi has shown a significant impact on shoot and root biomass, particularly in cool-season grasses like Agrostis stolonifera and Poa annua. Under phosphorus sufficiency (>35 ppm), its application increased biomass and phosphorus concentration in the plant and substrate, while under phosphorus deficiency (<five ppm), it reduced shoot and root development (Dempsey et al., 2022). These results reflect a hormetic response to Phi, where high doses can induce stress and reduce photosynthesis, while low doses stabilize key molecules like chlorophyll (Bojović & Stojanović, 2006; Cha-Um & Kirdmanee, 2008).

In strawberry (Fragaria x ananassa), lisianthus (Eustoma grandiflorum), lettuce (Lactuca sativa), and chard (Beta vulgaris var. cicla), Phi has improved quality and yield, increasing biomass, chlorophyll concentration, and fruit quality (Estrada-Ortiz et al., 2013; Estrada-Ortiz et al., 2016; Torres-Flores et al., 2018). In Brassica species, such as Brassica campestris and Brassica juncea, its application increased dry biomass weight and leaf area (Trejo-Téllez et al., 2019).

In hydroponic crops like spinach, doses of Phi combined with different phosphate levels increased fresh root weight under phosphorus deficiency (Thao et al., 2008). In potato (Solanum tuberosum) and tomato (Solanum lycopersicum), Phi improved flowering and postharvest quality (Lovatt & Mikkelsen, 2006).

The increase in fresh and dry biomass weight in this study is also related to the effects of Phi on photosynthesis, including enhanced photosynthetic activity, hydraulic conductance, electron transport chain efficiency, and aquaporin expression, which are fundamental under water stress conditions (Akram et al., 2018). The ability of Phi to stabilize chlorophyll and counteract the production of reactive oxygen species (ROS) also plays a crucial role in stress tolerance (Akram et al., 2018; Márquez-García et al., 2011). In our study, the 0.3 mM Phi dose significantly increased total chlorophyll concentrations and the chlorophyll a/b ratio, even under osmotic stress.

Chlorophyll content is another critical parameter affected by Phi. In sugarcane, the 0.3 mM dose significantly increased chlorophyll a, b, and total concentrations, both under conventional conditions and osmotic stress (Figs. 4A, 4B, 4C, 4D). This increase is related to the stability of photosynthetic pigments, which favor greater light absorption and, consequently, higher photosynthetic rates (Savvas & Ntatsi, 2015).

In Arabidopsis thaliana, Phi at 30% to 50% of total phosphorus caused changes in leaf coloration, correlated with variations in chlorophyll concentration (Ticconi, Delatorre & Abel, 2001). Similarly, in wild strawberry (Fragaria vesca), Phi promoted an increase in total chlorophyll during fruiting (Blanke, 2002).

The ability of Phi to stabilize chlorophyll and counteract the production of reactive oxygen species (ROS) also plays a crucial role in stress tolerance (Kellos et al., 2008; Márquez-García et al., 2011). In our study, the 0.3 mM Phi dose significantly increased total chlorophyll concentrations and the chlorophyll a/b ratio, even under osmotic stress. Although osmotic stress tends to reduce photosynthetic efficiency and sugar content in sink organs, our results show that Phi mitigates these negative effects, promoting the mobilization of photoassimilates.

Phi also affects the accumulation of essential biomolecules in plants. For example, it promotes the increase of free amino acids such as aspartic acid, glutamine, and asparagine, which are essential for nitrogen metabolism and adaptation to abiotic stress (Estrada-Ortiz et al., 2011; Ouellette et al., 2017). In this study, the addition of 0.3 mM Phi significantly increased amino acid concentrations in sugarcane seedlings under normal conditions, while under osmotic stress induced by 10% PEG, it partially mitigated adverse effects (Fig. 5A).

On the other hand, total soluble sugars, key indicators of nutritional status, also increased with Phi application. This effect was observed in strawberry during flowering and fruiting stages (Estrada-Ortiz et al., 2011), and in this study, under osmotic stress, Phi helped maintain high levels of soluble sugars by improving photosynthetic activity and photoassimilate mobilization (Zulfiqar, Akram & Ashraf, 2020).

Proline, a marker of drought tolerance, also showed a significant increase under osmotic stress with the application of 0.3 mM Phi. This amino acid acts as an antioxidant and membrane stabilizer, protecting seedlings against cellular damage induced by water stress (Hernández-Pérez et al., 2021). The application of Phi also improved the quality of various crops, such as strawberries, lisianthus, and lettuce, and its potential as a biostimulant is well-documented in different agricultural contexts (Estrada-Ortiz et al., 2016; Ricci et al., 2019).

Phi plays an important role in plant stress tolerance, such as drought. Under these conditions, compounds like proline, which stabilize membranes and proteins, increase in response to Phi (Hsu & Kao, 2003; Nath et al., 2018). Similarly, the ability of Phi to stimulate secondary metabolic processes, such as the accumulation of sugars and amino acids, strengthens osmotic stress resistance and maintains crop quality (Ricci et al., 2019).

Phytohormones play a key role in regulating hormonal signaling pathways related to stress responses in plants. Abscisic acid (ABA) has a primary role in stress responses in sugarcane, integrating physiological signals (stomatal closure, root growth), biochemical signals such as ROS modulation, and molecular signals when exposed to water stress, including an increase in ABA quantity and expression (Ferreira et al., 2017; Nong et al., 2024). In various crops, ABA regulates stomatal closure and water accumulation in tissues to counteract water deficit (Saradadevi, Palta & Siddique, 2017).

The interaction of phosphite with reactive oxygen species (ROS) and calcium-dependent pathways (Lim et al., 2013) indicates a connection between hormonal signaling and redox mechanisms in the cell. This finding could partly explain the rapid response of phosphite-treated seedlings to stress conditions. However, it remains necessary to determine whether this response is transient or if it generates lasting epigenetic effects that could influence long-term plant adaptation.

In Arabidopsis thaliana, phosphite (Phi) is absorbed via high-affinity phosphate transporters (PHT1) and regulates lipid remodeling genes (SQD2, NMT3) and ubiquitination pathways (PUB35, C3HC4). Systemic responses in leaves involve components of the PHO regulon (IPS1, SPX1, PHO1;H1). Phi induces PHO1 in roots, linked to vacuolar phosphate retention, revealing a stratified transcriptional regulation dependent on transporter specificity and temporal accumulation dynamics (Jost et al., 2015).

In transgenic plants, Phi disrupts the phosphate starvation response (PSR) by inhibiting ATPases and phosphofructokinases, thereby blocking Pi stress signaling. Concurrently, it modulates hormonal pathways (abscisic acid (ABA), salicylic acid (SA), jasmonic acid (JA)) and stress-related metabolites, enhancing anthocyanin synthesis and antioxidant enzymes (superoxide dismutase (SOD), catalase (CAT)), which improves tolerance to abiotic and biotic stress (Li et al., 2022). Furthermore, expression of the PtxD gene in tobacco (Nicotiana tabacum) enables the oxidation of Phi to phosphate (Pi), providing phosphorus nutrition while suppressing the growth of weeds such as Brachypodium distachyon, Alexander grass (Brachiaria plantaginea), morning-glory (Ipomoea purpurea), and smooth pigweed (Amaranthus hybridus). This demonstrates that foliar Phi application can effectively inhibit weeds or alleviate abiotic stress symptoms (López-Arredondo & Herrera-Estrella, 2012).

Another relevant aspect is the energy regulation associated with the activation of these pathways. The selective activation of defense genes without inducing excessive energy expenditure (Xiong, Schumaker & Zhu, 2002) suggests that phosphite optimizes plant responses without compromising growth. This balance is crucial for implementing phosphite-based strategies in agricultural systems, as it allows for improved crop resistance without affecting yield.

The separation of treatments observed in the PCA confirms that Phi significantly modulates the physiological and biochemical responses of sugarcane seedlings. The clustering of treatments with 0.3 mM and 0.5 mM Phi without PEG in the upper right quadrant suggests a strong relationship between Phi application and increased plant growth, chlorophyll accumulation, and improved osmoprotective metabolites, such as proline and soluble sugars. These results align with previous studies in Arabidopsis thaliana and Solanum tuberosum, where Phi application stimulated the expression of genes associated with primary metabolism and phosphate transport, favoring plant development under stress conditions (Jost et al., 2015; Datta et al., 2024).

The distribution pattern in the heatmap suggests that Phi not only improves biomass and photosynthetic efficiency under normal conditions but also modulates key biochemical responses under osmotic stress. The accumulation of proline and soluble sugars in treatments with 0.3 mM Phi + 10% PEG indicates that this compound acts as an inducer of stress tolerance mechanisms, facilitating cellular homeostasis and membrane stability under adverse conditions (Van den Ende, 2014; Zulfiqar, Akram & Ashraf, 2020).

Conclusions

The application of phosphite (Phi), both under conventional conditions and in the presence of osmotic stress induced by 10% PEG, produced significant modifications in morphometric and metabolic parameters in sugarcane seedlings (variety CP 72-2086) cultivated in vitro.

Under conventional conditions (without osmotic stress), Phi stimulated plant growth by increasing leaf length, width, and number, as well as shoot number. Additionally, it promoted the accumulation of chlorophyll a and b, soluble sugars, and essential amino acids. These responses reflect an optimization of primary metabolism, possibly mediated by the interaction of Phi with physiological and biochemical pathways related to phosphorus metabolism and the transport of essential nutrients.

Under osmotic stress (10% PEG), Phi acted as a mitigating agent, improving total chlorophyll concentration, the chlorophyll a/b ratio, and proline content. This osmoprotective amino acid plays a crucial role in stabilizing membranes and proteins, as well as reducing the negative effects of water stress, such as ROS production. These responses highlight the ability of Phi to modulate abiotic stress tolerance mechanisms through the induction of key metabolites that enhance seedling resilience.

Therefore, Phi positions itself as a promising biostimulant for the in vitro micropropagation of sugarcane, as it not only enhances growth under conventional conditions but also increases tolerance to osmotic stress. This finding has significant implications for countries like Mexico, where agricultural systems face challenges related to climate change, water scarcity, and the need to increase crop production sustainably.

The use of Phi could contribute to obtaining healthy and vigorous seedlings in early stages, improving the productivity and efficiency of agricultural systems. These results open the possibility of exploring its application in other crops and agricultural conditions, promoting integrated strategies for sustainable management and resilience in the face of current and future environmental pressures.

Supplemental Information

Supplemental Information 1 Raw data

Supplemental Information 2 Script R

Additional Information and Declarations

Competing Interests

Author Contributions

Data Availability

The authors declare there are no competing interests.

Jennifer Martínez-Ballesteros conceived and designed the experiments, performed the experiments, analyzed the data, prepared figures and/or tables, authored or reviewed drafts of the article, and approved the final draft.

Karina P. Bañuelos-Hernández conceived and designed the experiments, authored or reviewed drafts of the article, and approved the final draft.

Daniel A. Rodríguez-Lagunes conceived and designed the experiments, authored or reviewed drafts of the article, and approved the final draft.

Juan V. Hidalgo-Contreras conceived and designed the experiments, analyzed the data, authored or reviewed drafts of the article, and approved the final draft.

Miriam C. Pastelín-Solano conceived and designed the experiments, authored or reviewed drafts of the article, and approved the final draft.

Guadalupe Vivar-Vera conceived and designed the experiments, authored or reviewed drafts of the article, and approved the final draft.

Javier E. Bulbarela-Marini conceived and designed the experiments, authored or reviewed drafts of the article, and approved the final draft.

Odon Castañeda-Castro conceived and designed the experiments, performed the experiments, prepared figures and/or tables, authored or reviewed drafts of the article, and approved the final draft.

The following information was supplied regarding data availability:

The R script with ANOVAs and principal component analysis are available in the Supplementary Files.

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
