# Peer review of "Phosphite effects on sugarcane growth and biochemicals under in vitro osmotic stress"

_PeerJ, doi:10.7717/peerj.19763_

## Round 0.1 · original submission · Minor Revisions

The manuscript presents relevant findings on the role of phosphite (Phi) in mitigating osmotic stress in sugarcane under in vitro conditions. Both reviewers agree on the scientific interest of the study and recommend minor revisions before publication.

The experimental design and statistical approach need clarification. The manuscript lacks explicit statistical data showing the interaction effects—this should be either included or discussed, since PCA alone cannot validate interaction.

Figures require improvements: Figures 2–7 should display mean values, and the statistical letters need better alignment. In Figure 4B, an error bar appears incomplete. In Figure 6, the bar for 0% PEG in the proline analysis is missing. In Figure 9, axes should be more informative with treatment names to improve reader understanding. Reviewer 2 also noted that Figure 1 may show duplicated images (C and D); this should be verified and corrected if necessary.
There are also inconsistencies between the text and figures. For instance, the claim in line 256 that all Phi treatments differ significantly from the control is not supported by the data in Figure 6. All statements regarding statistical differences should be carefully reviewed and aligned with the data.

Additional context is needed in the Introduction, especially regarding the sugarcane cultivar CP 72-2086 (e.g., agronomic relevance, geographic use, yield potential) and previous studies supporting the use of 10% PEG under sterile conditions. Reviewer 2 also suggests enriching the final paragraph with references on Phi-induced drought/osmotic tolerance.

From a stylistic standpoint, the Results section can be streamlined. Repetition of small numeric details should be avoided, as figures are self-explanatory. Focus on key findings. Throughout the text, grammatical and formatting corrections are also needed. Finally, adding a brief paragraph in the Discussion about potential mechanisms of Phi in osmotic stress response would add value.

All reviewer suggestions should be addressed in the revised version. Additionally, the title suggestion proposed by Reviewer 1 is appropriate and may be adopted to better reflect the focus of the manuscript.

Reviewer 1 ·

Basic reporting

1. In the introduction section, there is a lack of information on the sugarcane variety CP 72-2086 used in this experiment. The author should provide basic information such as planting area, yield, etc., including the reasons for choosing this sugarcane variety for the experiment.
2. The authors should cite sugarcane studies that support using 10% PEG for inducing stress under sterile conditions, as other concentrations have been used.
3. For enhanced clarity, Figures 2 through 7 should include the mean values of the experimental results, facilitating reader comprehension.
4. A discrepancy exists between the text and Figure 6. The proline concentration data for the 0.3 mM phosphite treatment, as detailed in line 252, is absent from the figure.
5. To enhance the understandability of Figure 9, the authors should include treatment identifiers on both axes, allowing readers to follow the experimental results more easily.

Experimental design

1. Why did the authors choose a CRD design with two-way ANOVA for this research, rather than a factorial or split-plot in CRD design, which could better explain the interaction between phosphite and stress conditions? The authors can adjust the analysis to be factorial in CRD to see the influence of phosphite and PEG concentration levels and their interaction.

Validity of the findings

1. The authors should carefully review their statistical interpretations. For example, line 256 states, "The remaining treatments with Phi and Phi + 10% PEG were also statistically superior to the control (Figure 6)." However, Figure 6 shows that Phi 0.5 mM + 10% PEG is not significantly different from the control.
2. The authors did not present the statistical analysis results showing the interaction between Phi and osmotic stress in the experimental results. Although the authors used PCA to indicate the interaction trend between Phi and osmotic stress, PCA could not identify the influence of the interaction.
3. Research showed that the application of Phosphite (Phi) affected sugarcane's growth and biochemical indicators under normal and osmotic stress conditions induced by Polyethylene Glycol (PEG). The experimental results showed that Phi could help reduce the adverse effects of osmotic stress and promote sugarcane growth.
4. An appropriate control group (i.e., no Phi and no PEG) was used to compare the effects of Phi and PEG, which provided reliable data.
5. To increase the interest of the research article, the author could add a review section on the molecular mechanism of Phi to provide a clearer understanding of why Phi can help reduce the adverse effects of Osmotic stress.
6. This study demonstrates the effects of phosphite (Phi) on growth, accumulation of active metabolites, and response to osmotic stress in sugarcane. It identifies the mechanisms involved and their potential application for sustainable yield enhancement. Importantly, this study investigated the effects of Phi under in vitro conditions that simulate osmotic stress in sugarcane, which have been limitedly studied, and provides new information that is important for the development of future stress management strategies in sugarcane.

Additional comments

The author may revise the title to "Phosphite Effects on Sugarcane Growth and Biochemicals Under In Vitro Osmotic Stress" to make it more concise.

·

Basic reporting

The present study reports the in vitro analysis of the effects of phosphite biostimulant on growth and biochemical parameters of sugarcane plantlets subjected to osmotic stress induced by different doses of polyethylene glycol. Overall, the English language used in the manuscript is clear and technically and scientifically sound. The introduction part, though comprehensively written seems a bit short. In the last paragraph, if possible, a few more references/previous reports regarding the role of phosphite in osmotic/drought stress tolerance in sugarcane and other crops should be included. The research question is well-defined, relevant and clearly states the knowledge gap and the strategy to address the questions.

Experimental design

The experimental design is well-written with all necessary investigations and relevant references. The methods are described with sufficient details and proper replications. Moreover, sufficient statistical tools have been used for analysis of data, which further ensures clarity, originality and quality of the obtained data. Data has been presented in a scientifically sound manner.

Validity of the findings

The discussion and conclusion parts are well-written and fully support and link the results with the study questions. Despite, the overall good quality of the data, some minor changes will further improve the manuscript.

Additional comments

The following points should be considered.
1) In introduction, line#66, please correct (vitro multiplication).
2) In the result section throughout, minor details of numerical data from figures are mentioned. There is no need to mention minor details. Just focus on significant findings in each figure. Figures are self-explanatory, so no need of detailed description in the result section. Readers are only interested in major findings.
3) In figure 1, it seems the C and D are same plants, though these are different treatments. Please check and make corrections.
4) In figures 2-7, the letters above the error bars should be placed closer. The gap should be reduced.
5) In figure 4B (0.3 mM), please check the error bar is incomplete.
6) In figure 6, the proline data bar at 0 PEG is missing.
7) Please check the entire manuscript for grammatical and formatting errors.

---

## Round 0.2 · Minor Revisions

I agree with the points raised; authors just need to satisfy reviewer #1 claims

Reviewer 1 ·

Basic reporting

The manuscript meets all four standards. It is written in clear English and adheres to the journal format, providing sufficient background and context with appropriate references. The figures are relevant and well-described.

Experimental design

The manuscript addresses an original research question within the journal's scope, clearly defining the aims and their relevance. The investigation is thorough, conducted to scientific and ethical standards. Methods are described in sufficient detail.

Validity of the findings

The discussion and conclusion sections are clearly articulated and effectively connect the findings to the research questions. The overall quality of this manuscript is good; however, there is a minor typo on line 203, where an uncited reference appears after the word "pheatmap." Please add the appropriate citation to improve accuracy.

·

Basic reporting

the suggested changes were incorporated

Experimental design

the suggested changes were incorporated

Validity of the findings

the suggested changes were incorporated

Additional comments

the suggested changes were incorporated

---

## Round 0.3 · accepted · Accept

The reviewer who requested minor revisions provided the previous assessment, and I confirm that the authors have addressed all comments. I have reviewed the revised version and consider it ready for publication.

Reviewer 1 ·

Basic reporting

The manuscript meets all four standards. It is written in clear English and adheres to the journal format, providing sufficient background and context with appropriate references. The figures are relevant and well-described.

Experimental design

The manuscript addresses an original research question within the journal's scope, clearly defining the aims and their relevance. The investigation is thorough, conducted to scientific and ethical standards. Methods are described in sufficient detail.

Validity of the findings

The discussion and conclusion sections are clearly articulated and effectively connect the findings to the research questions. The overall quality of this manuscript is good.

Additional comments

-